# Sticking to the Subject: Multifunctionality in Microbial Adhesins

**DOI:** 10.3390/jof9040419

**Published:** 2023-03-29

**Authors:** Peter N. Lipke, Peleg Ragonis-Bachar

**Affiliations:** 1Biology Department, Brooklyn College of the City University of New York, Brooklyn, NY 11215, USA; 2Department of Biology, Technion-Israel Institute of Technology, Haifa 3200003, Israel

**Keywords:** bacterial cell wall, fungal cell wall, *k*
_off_, multidomain protein, moonlighting protein, functional amyloid

## Abstract

Bacterial and fungal adhesins mediate microbial aggregation, biofilm formation, and adhesion to host. We divide these proteins into two major classes: professional adhesins and moonlighting adhesins that have a non-adhesive activity that is evolutionarily conserved. A fundamental difference between the two classes is the dissociation rate. Whereas moonlighters, including cytoplasmic enzymes and chaperones, can bind with high affinity, they usually dissociate quickly. Professional adhesins often have unusually long dissociation rates: minutes or hours. Each adhesin has at least three activities: cell surface association, binding to a ligand or adhesive partner protein, and as a microbial surface pattern for host recognition. We briefly discuss *Bacillus subtilis* TasA, pilin adhesins, gram positive MSCRAMMs, and yeast mating adhesins, lectins and flocculins, and *Candida* Awp and Als families. For these professional adhesins, multiple activities include binding to diverse ligands and binding partners, assembly into molecular complexes, maintenance of cell wall integrity, signaling for cellular differentiation in biofilms and in mating, surface amyloid formation, and anchorage of moonlighting adhesins. We summarize the structural features that lead to these diverse activities. We conclude that adhesins resemble other proteins with multiple activities, but they have unique structural features to facilitate multifunctionality.

## 1. Introduction

Textbooks often state that each protein has a primary function mediated by a binding region or active site. This characterization may be a historical relict of the “One gene, one enzyme” hypothesis. However, as the theme of this journal issue makes clear, in many cases this is not correct. An enzyme’s active site is a substrate-binding region shaped by sequence and fold, and often consists of <10% of the amino acids in a polypeptide. The idea of a single function implies that the binding region is the kernel of a protein, and so the rest of the residues merely provide a scaffold to ensure correctly folded geometry of the active site. However, we know that any protein fold can be established by myriad different sequences that generate the same protein geometry [1,2]. This diversity means that the 90% of residues that are not in the active site binding region can and do evolve while preserving the protein fold. Often, these regions of the protein fulfill other functions and activities while maintaining a specific conformation. The microbial adhesins show many of these diverse characteristics. 

Microbial adhesins have multiple functions, and these activities are all evolved and selected. Adhesins can act as enzymes [3], as assembly scaffolds and components of complex nano-machines [4]. Sometimes, these activities are called secondary because they were discovered secondarily. For instance, microbial type IV pili were first called adhesive, but we now know that they function in gliding motility [5]. In contrast, phosphoglycerate kinase has its well-known enzymatic activity, but in fungi it also moonlights as an extracellular adhesin [6]. The current convention is that a protein should be named for its most conserved activity, rather than the first activity discovered.

## 2. Multiple Activities of Microbial Adhesins

Multiple activities and multiple binding sites are common in microbial adhesins. Here, we focus on some similarities and emerging trends. This essay is thus a ‘meta-review’ covering some key ideas and concepts. Most of the references are therefore subject-specific reviews, rather than the primary literature. 

### 2.1. What Is an Adhesin?

This question has a simple answer: a cell surface protein that binds one cell to another or binds a cell to a substrate. This definition encompasses two types of proteins: professional, dedicated adhesins and ‘moonlighting’ proteins that are displayed on a cell surface and happen to bind to a partner on another cell. Both types of adhesin are multifunctional and are found on the surface of both prokaryotic and eukaryotic microbes. For microbes, many of these adhesins are involved in microbe-host in teractions, in biofilm formation and maintenance, or both. This essay concentrates on those adhesins that have cell–cell interaction activity. Nevertheless, many of these adhesins also bind microbes to biotic and abiotic surfaces. 

### 2.2. Biofilms

Biofilms are often considered to be the natural state of microbial growth and maintenance. They are organized communities in which cells are differentiated and adapted to specific metabolic and adhesive roles. Cells in both bacterial and fungal biofilms are also characterized by high resistance to antibiotics and antifungals [7,8]. Biofilms are also resistant to nematode predation, a property conferred by the adhesins [9,10]. Biofilms include at least four different morphologies: amorphous layers on surfaces; surface layers with perpendicular towers or mushroom-shaped constructs; well-organized mats of cells with regular wrinkles and crenellations; and pellicles, biofilms formed at air-water interfaces [7,11]. Within each biofilm morphology, individual cells have different specialized roles dependent on specific patterns of gene expression [12,13,14].

### 2.3. Multiple Activities of Microbial Adhesins

Minimally, each adhesin has three activities (Figure 1):Its ability to recognize and bind a ligand or other binding partner. This binding must last long enough for the adhesion to be biologically relevant; often this means that the adhesive bond lasts for hours or even years [15];Its attachment to the surface of the expressing cell. Attachment can be through covalent binding to the cell wall, membrane embedment, or association with another wall-attached protein [16];Its activity as a surface marker: This activity is crucial in microbe-host interactions, and many adhesins are also immune modulators. Adhesin structures are bound by immune effectors and by other cell signal receptors. This binding leads to the modulation of host responses. In addition, biofilm adhesins mediate microbe–microbe associations, which induce changes in microbial cellular physiology [12,13,17]. This activity can be due to direct signaling between adhesins, or indirect due to adhesin-induced long-term increases in population density [18,19].

### 2.4. Time and Adhesin Activity

A simple definition of an adhesin is a protein that mediates direct binding of one cell to another. By convention, the definition excludes many intercellular signaling complexes that induce cellular differentiation or alter cellular activity within a few seconds. The timing is somewhat flexible: for instance, lectin binding is brief and allows lymphocytes to roll along the endothelium as successive molecules bind and dissociate [20]; on the other hand, the adhesins that mediate bacterial mating are active for 20 min to an hour [21]. There are also instances where continued adhesion leads to continuing signaling that alters cell fate: in yeast mating, responses to sex pheromone include changes in gene expression after a few seconds of exposure, but the ability to elicit a mating process requires at least an hour of response to high pheromone concentration, and this response requires continued cell–cell adhesion [22,23]. Similarly, T cells respond rapidly to stimulation by antigen presenting cells (APCs), but continued stimulation for hours to days facilitates maximal proliferation, and can also alter cellular differentiation [24,25,26]. Thus, cell-cell adhesion systems evolve to match the needed kinetics and the duration of cell-cell binding. Among microbes, cell adhesion can last seconds to minutes to entire lifetimes (especially for cells in persistent biofilms and in lichens) [15,27,28]. The requirements for temporary or permanent contacts are achieved through variations in dissociation rates and in avidity. (Avidity is the concept that for cells attached through multiple bonds, the probability of separation of the cells decreases exponentially with a linear increase in number of intercellular bonds.) Thus, frequent characteristics of professional adhesins include low dissociation rates and high avidity through multivalence, often through adhesin clustering on the cell surface [29,30]. It follows that proteins that are moonlighting as adhesins are likely to have higher dissociation rates, be clustered less often, and therefore mediate cell-cell interactions that are short-lived [31,32].

### 2.5. Adhesion Assays

The question, therefore, is how can we demonstrate that a given protein is an adhesin? The gold standard assays show that a specific protein is both necessary and sufficient for the adhesion of cells to each other or to a substrate. Genetically, these properties are often established by showing that adhesion is lost when a specific gene is mutated (necessity demonstrated by loss of function, LoF), and, conversely, that expression of the protein in a heterologous system confers adhesion activity (sufficiency shown by gain of function) [33]. In practice, LoF mutations are often used to identify putative adhesins, but they can be confounded by other activities of the proteins (for instance, several putative adhesins are transcription factors, or inducers of transcription regulation, and deletions cause failure to express other cell-surface adhesins [34]). Gain of function assays are less frequent in the literature. These usually involve exogenous expression of adhesins that render a non-adherent cell adherent, or less frequently, a demonstration that a specific protein confers adhesion activity when attached to synthetic beads. Cell surface localization is also necessary, but it is not sufficient, because many cytoplasmic proteins can be unconventionally secreted [35]. Of course, each adhesion assay is dependent on the specific surface or specific cells the adhesin is binding [16].

### 2.6. Characteristics of Adhesin Binding

#### 2.6.1. Adhesin Ligands and Partners

There are many ways that adhesins bind to target proteins. In fact, the standard ligand receptor terminology is often inappropriate when both binders are adhesins that contribute large binding surfaces. Thus, Figure 2 shows several examples. In (A), a protein with a well-folded domain incorporating a well-defined pocket binds to a small segment of a heterologous adhesin. This cartoon is like a classic receptor-ligand model, and an example is the *Saccharomyces cerevisiae* mating adhesin alpha-agglutinin, which binds the C-terminal residues of its partner, **a**-agglutinin subunit Aga2. The existence of such binding pockets is, of course, analogous to active sites on most enzymes. In (B) and (C), heterologous adhesins bind through interactions of larger portions of the domain surfaces. Finally, in (D), there is extensive interaction between two identical adhesins to show homophilic binding.

#### 2.6.2. Adhesin Bonding

The vast majority of adhesin interactions are non-covalent. These events include all of the modes of protein-protein binding including Van der Waals, polar, and ionic interactions, as well as the hydrophobic effect between complementary surfaces. An exception is the *Candida albicans* adhesin Hwp1, one of a class of adhesins covalently bound to fungal cell surface through glycosyl phosphatidyl inositol (GPI) links to cell wall glucans. Hwp1 is a substrate for transglutaminase on mammalian cell surface, and the enzyme crosslinks Gln residues in the N-terminal region of Hwp1 to Lys residues on the surface of mammalian keratinocytes [36,37]. It is also possible that disulfide exchange can form intercellular disulfides in yeast mating. Because it is possible that adhesins themselves promote these disulfide exchange reactions, such activity represents the multifunctionality of adhesins. However, this speculation has never been tested.

In summary, multiple activities and multiple binding sites are common in microbial adhesins. Here, we focus on some similarities and emerging trends.

## 3. Intracellular Proteins That Moonlight as Adhesins

In both bacteria and fungi, highly expressed cytoplasmic and mitochondrial proteins have often been identified as adhesins. Among those most often identified are the glycolytic enzymes enolase and glyceraldehyde phosphate dehydrogenase (GAPDH), along with cytoplasmic chaperones. These proteins are consistently identified as cell wall components, despite their lack of secretion signals. In bacteria the mechanism of externalization is still not clear [38]. In fungi they are unconventionally secreted through the endosomal/vacuolar system [35]. Such moonlighters have often been identified as receptors for plasma and extracellular matrix (ECM) proteins, and their fungal occurrence, activities, and suitability as drug targets has recently been comprehensively reviewed [34]. Table 1 in that paper is an excellent recent and comprehensive summary and is illustrated in Figure 3. Of note, *C. albicans* enolase binds to multiple plasma proteins, ECM proteins, and protein components of antifungal NETS in neutrophil granules, and its deletion affects hyphal morphology as well as virulence. *C. albicans* enolase is also a transglutaminase, and this activity is critical for its roles in hyphal morphology and virulence [3,39]. Enolase is anchored in the cell wall through binding to the adhesin Als3 [40]. Thus, there is a moonlight conversation between the adhesins Als3, Hwp1, Eno1, and mammalian transglutaminases.

## 4. Multifunctional Bacterial Adhesins

Current lists include about 1800 genera of bacteria, and each species expresses several adhesins that have multiple binding partners or ligands. We mention only a tiny sample of bacterial adhesins in three different phyla, and with different cellular locations and different diversity of functions.

### 4.1. Bacillus subtilis TasA

This ~30 kD protein has a wide variety of effects. It was discovered as a component of spore coats. There are at least two forms of the protein: a soluble monomeric α/β globular form that forms amyloid-like fibers in vivo, which, however, lack canonical cross-β structure [19]. Fibrous TasA is a major component of biofilm matrix, where it contributes to biofilm wrinkling (thought to be a strategy for making nutrient channels within the biofilm [41]. Unsurprisingly, TasA determines biofilm hydrophobicity and cohesion under stress. It also contributes to antibiotic resistance of biofilms. However, TasA also acts intracellularly upstream of SinR to regulate cell fate. Mutant *tasA* cells form biofilms with fewer motile cells and more cells that excrete extracellular matrix components [42]. Accordingly, transcriptomics shows that *TasA* contributes to the regulation of genes involved in motility and secondary metabolism. The mechanisms behind these non-adhesive functions are only now being addressed. Thus, TasA is a truly multifunctional protein with effects on cell adhesion, substrate adhesion, biofilm structure, cell fate, and physiology. The most similar fungal adhesin is the multifunctional *Candida albicans* Int1 described in Section 6.

### 4.2. Pili Adhesins

Bacterial pili and fimbriae often have adhesion as a primary function [43,44]. The adhesins are the tip subunits in some cases, or in other cases adhesion activity is a property of the major pilin structural subunits themselves. The curli pili of gram-negative bacteria assemble in amyloid-like structures, and greatly contribute to biofilm structure and adhesiveness and antibiotic resistance. There is extensive literature on the biosynthesis, assembly, and consequences of these structures [45,46]. Among these, amyloid-like curli are now being used as a platform for synthetic biology structures with multiple adhesion specificities [47]. Pili adhesins have a similar way of binding partners with fungal adhesins, and amyloid-like aggregation is now well known in fungal adhesins (Section 5, Section 6, Section 7, Section 8 and Section 9 below) [29].

### 4.3. MSCRAMMS

Gram-positive cocci like *Staphylococcus* and *Streptococcus* express MSCRAMMS (microbial surface components recognizing adhesive matrix molecules) [48]. These adhesins are cross-linked through their C-terminal residues to the wall peptidoglycan by transpeptidylation during wall synthesis [49]. Various MSCRAMMS bind to a large variety of mammalian ECM components including fibrinogen, collagens, fibronectin, cytokeratins, complement factors, and others. Most MSCRAMMS are 500–1000 amino acids. They have tandem Ig-like β-sandwich domains near the N-terminus. A primary binding site is in the linear cleft or trench between the domains. Ligand peptides dock in the cleft through sidechain interactions. Then, an unstructured segment of the protein immediately C-terminal to the second Ig-like domain locks the ligand in place by crossing on top of it. This action forms a ‘latch’ onto the first Ig-like domain by constituting an additional β-strand at the edge of one β-sheet (Figure 2 in [48]). This dock, lock, and latch mechanism (DLL) results in an extremely strong adhesive bond. In the collagen-binding MSCRAMM CNA from *Staph. aureus*, the Ig-like domains are connected by a flexible linker. The two domains then close around the fiber, and the latch peptide adds stability by β-strand addition to domain I, as in DLL binding [48]. Most MSCRAMMS also have secondary binding sites for other ligands as well, so that they can bridge between structures and cells. The MSCRAMMS have properties highly similar to the Als adhesins of the fungus *Candida albicans* (see below).

## 5. General Characteristics of Fungal Adhesins

Each fungal adhesin has different specificity for binding, as well as a characteristic profile for expression time in the growth cycle in the various fungal morphs and growth phases. Nevertheless, they share certain molecular features, including heavy glycosylation, hydrophobicity, secretion signals, C-terminal modified glycosyl phosphatidyl inositol (GPI) anchors covalently bound to cell wall glucans, amyloid core sequences, and often Cys-rich motifs and dibasic motifs [16]. In a few cases, some of the resulting adhesion partners have been identified. Simple examples include the *S. cerevisiae* Flo adhesins. These adhesins facilitate formation and maintenance mats, a type of biofilm that looks like a large, structured colony on an agar plate [9,50,51,52]. The *FLO1* family encodes 3 adhesins (see Figure 3 in [29]), and each is a lectin with a glycan binding site specific for α-mannosyl or α-glucosyl residues [53]. The binding has moderate affinity, and is non-specific, considering the prevalence of these residues on fungal and animal cell surfaces. In addition, each Flo lectin has multiple amyloid core sequences that can form homotypic cross-β structures on cell surfaces to cluster the adhesins and increase avidity of the interactions. The formation of cross-β aggregates increases the strength of adhesive bonds. The adhesin Flo11 is not homologous to the Flo lectins and does not have glycan binding activity. Instead, its interactions include homotypic association between Trp residues arrayed in a ring around the surface of the well-folded N-terminal domain. The Trp indole and Tyr phenol rings stack to generate electron delocalization similar to that of base stacking in a DNA double helix. Trp and Tyr residues also contribute hydrophobic effect interactions [54]. In addition, there is formation of cross-β mediated cell surface aggregates, and probably also cross-β cell-to-cell bonds, similar to the *C. albicans* Als adhesins. These bonds are necessary for biofilm integrity and resistance to predation [52,55]. In addition, in yeasts that make sherry wine, the hydrophobicity of Flo11-mediated associations causes the yeast to form a biofilm pellicle at the air–water interface on top of the fermentation [56]. The conclusion is that almost all fungal adhesins have at least two binding modes, as well as several cellular functions.

## 6. *Candida albicans* Int1

This protein is large and highly multifunctional. Although its structure is largely unknown, it has low homology to mammalian integrins, and includes a region similar to the *Saccharomyces cerevisiae* Bud4 GTPase. Accordingly, it has intracellular functions including interaction with septin ring proteins. Its roles include ploidy stabilization, bud site selection, and in hyphal morphogenesis under some conditions [57,58]. A poly Gln sequence at the C-terminal is characteristic of proteins that regulate cellular responses through the formation of reversible membrane-less RNA-protein aggregates [59]. Despite its lack of a signal sequence, it can be detected on the cell surface, so it is presumably unconventionally secreted [35]. Int1 mediates binding to epithelial cells as well as binding of complement protein iC3b [57,60]. Thus, Int1 has evolved to function in many key features of cellular morphogenesis, and its adhesin activity is perhaps moonlighting.

## 7. Multiple Functions of Yeast Mating Adhesins

The *S. cerevisiae* mating process illustrates the multiple roles for adhesins (Figure 4). The classic adhesin Sag1 expressed by mating type α (*MATα*) cells binds to the C-terminal residues of Aga2, which is expressed only in mating type **a** (*MAT***a**) cells [23,61,62]. By homology arguments, this binding is similar to the *Candida albicans* Als adhesin family [63], as discussed below. *S. cerevisiae* Aga2 is the only known Sag1 ligand in the biosphere; thus, Sag1 and Aga2 have only a single known activity. However, Aga2 is a 69-amino acid mannoprotein, and is anchored to the cell surface through disulfide bonding to Aga1, a GPI-crosslinked cell wall protein. In contrast, Aga1, which appears to be an unstructured mannoprotein, is expressed on the surface of both mating types [64]. In matings on solid substrates, Sag1 and Aga2 are not essential, and at least in these mating conditions cell adhesion is mediated by Aga1 binding to its distant homolog Fig2 [64]. Thus, Aga1 is at least a bifunctional adhesin. Fig2 can also bind to itself to facilitate mating on surfaces. Binding Aga1-Fig2 and Fig2-Fig2 binding require conserved Cys residues in WCPY and CX_4_C motifs in both partners. Thus, it is possible but untested that these adhesins form intermolecular disulfide bonds, and such bonds would constitute *intercellular* disulfide bonds, during mating. All of these adhesins are required for normal morphology of the zygotes [65]. These adhesins Aga1 and Fig2 thus illustrate adhesins with multiple binding partners, and Aga1 has the additional role in disulfide binding to Aga2.

### Multiple Functions through Indirect Consequences of Adhesion in Mating

In *S. cerevisiae,* mating is initiated by peptide sex pheromones [66]. Each mating type secretes a pheromone recognized by a G-protein-coupled receptor (GPCR) expressed by the opposite mating type. A MAPK cascade (homologous to mammalian mitogen-activated protein kinase cascades) then leads to increased synthesis and secretion of pheromones, receptors, and adhesins, which are all localized on a conically-shaped cellular mating extension. This co-localization maximizes mating response in at least three ways. First, the slow *k_off_* and high avidity of the adhesive bonds leaves the cells in intimate contact for the several hours needed to complete mating. Second, the local concentrations of pheromones are 100- to 1000-fold higher than in bulk medium, and this high concentration as well as long term exposure are necessary to induce morphological and physiological processes in mating, including cell wall remodeling, membrane fusion, and nuclear fusion [22,23]. Thirdly, the close apposition of the mating cells inhibits diffusion of the enzymes that remodel the cell walls. Therefore, these lytic enzymes act only between the mating cells, and prevent lysis at other loci on the cell surface. In the absence of adhesion, cells undergo mating-induced cell death, because the lytic enzymes diffuse and lyse cells at other sites [67]. Therefore, even though no signaling function has been ascribed to the mating adhesins, their action ensures productive matings through prolonged contact, increased localization of high pheromone concentrations, and localization of wall remodeling.

## 8. Multiple Binding Modes and Multiple Functions in *Candida adhesins*

Dozens of adhesins have been identified in *C. albicans* and *C. glabrata,* human commensals and opportunistic pathogens. *C. albicans* is the most frequent agent for human fungal infections and the cause of most fungal-induced mortality [68]. *C. glabrata* is more closely related to baker’s yeast *S. cerevisiae* than to *C. albicans*, and it is highly resistant to many antifungals. Therefore, *C. glabrata* is now the second most frequent fungal disease agent. Each of these fungi expresses specific adhesins differentially in different growth conditions and in different fungal morphs and phases. These adhesins share certain characteristics, including heavy glycosylation, hydrophobicity, GPI-mediated bonds to wall glucans, and amyloid core sequences [16]. In *C. glabrata*, two large adhesin families, *EPA* and *AWP*, have been well-studied [53,69]. The *EPA* family encodes many glycoprotein lectins that bind galactose and related saccharides in mammalian glycocalyx. In each adhesin, an N-terminal C-type lectin domain is attached to a long, Ser/Thr-rich glycosylated stalk and a modified GPI anchor covalently bonded to cell wall glucan. Epa family adhesins do not appear to use amyloid-mediated interactions, a characteristic of *S. cerevisiae* Flo family lectins. Therefore, *EPA* family lectins are likely to mediate short-lived interactions, like the lectins that mediate rolling of lymphocytes along the endothelium [20].

### 8.1. Candida glabrata Awp Family Adhesins

The Awp family of paralogous adhesins is multifunctional in that its members mediate fungal aggregation as well as adhesion to plastic, glass, mammalian cells [70]. Awp proteins have β-helical structure, i.e., a cylinder with β-strands perpendicular to the cylinder axis arranged helically (Figure 5). The cylinder is ~60 Å in length and 15 Å in diameter, so this structure gives large surface areas that can display diverse amino acid side chains that mediate interactions with many different binding partners. By an unknown mechanism, Awp2 also contributes to cell wall integrity.

### 8.2. C. albicans Adhesins

*C. albicans* expresses dozens of surface glycoproteins with adhesin activity [16]. Among these, Hwp1 was mentioned above as an adhesin that can function as a substrate for transglutaminase crosslinking to mammalian cells [36]. The best characterized adhesins are products of the *ALS* gene family.

## 9. *C. albicans* Als Family Adhesins

In *C. albicans* eight Als adhesins are encoded at separate loci, *ALS1* through *ALS7* and *ALS9.* These loci are usually heteroallelic (i.e., the diploid alleles have different sequences, so 16 different gene products are encoded) [71,72]. More diversity is generated by the ambiguity of the CUG codon, which is translated as Ser 97% of the time, and as Leu 3% of the time. CUG encodes 2–18 surface-exposed residues in different Als proteins [73]. Even at this low amount of ambiguity, many adhesin molecules transcribed from the same gene display non-identical peptide sequences; individual molecules also vary in glycosylation patterns. This heterogeneity generates adhesins with different binding characteristics. Nevertheless, all members of the gene family share the same basic architecture (Figure 6). Each adhesin has a tandem pair of N-terminal Ig-like invasin domains, and there is a peptide binding cleft at the interface between the domains. In each adhesin, there is a sequence-conserved amyloid core sequence immediately following this region. This sequence is initially folded at the interface between the highly conserved T region and the Ig-like domains [29]. C-terminal to the T region are 5–35 copies of a 36-amino acid tandem repeat with a three-stranded antiparallel β-sheet structure. The tandem repeats are followed by a flexible but unstructured stalk region of 600–1000 amino acids with 30–50% glycosylated residues, then the modified GPI anchor that mediates attachment to the cell wall glucan. Similar adhesins are encoded in many diverse yeasts, including *S. cerevisiae* α-agglutinin (Sag1).

### 9.1. Adhesion-Related Activities of Als Family Proteins

Steve Klotz and PNL were interested in how two homologous proteins could have such diverse binding specificities. Lipke’s group and collaborators had worked for two decades on *S. cerevisiae* Sag1, which binds only Aga2. Klotz and Nand Gaur had discovered that Als5 bound strongly to a wide variety of substrates, including mammalian ECM proteins. Many researchers have since learned that Als proteins have evolved to have myriad binding partners, and therefore they interact through a large catalog of molecular surfaces and noncovalent binding modes (Figure 6). These binding interactions are key in biofilm formation, so they have been called “social” [29]. On the other hand, they also mediate “antisocial” interactions: pathogenesis and immune escape [30,74,75,76].

The binding specificity of *S. cerevisiae* Sag1 is very strict: the only ligand is the Aga2 C-terminal sequence …TyrValPhe-COO^−^ [61,62,77], and this sequence is highly species-specific [78]. This specificity is apparently due to a constricted binding cavity between the two Ig-like invasin domains; the residues lining the binding site interact with these specific sidechains [61,79].

Contrastingly, in Als family adhesins, the interface between the two Ig-like invasin domains forms a shallow cleft with one end occupied by a conserved Lys residue. The amino group of the Lys forms an ion pair with C-terminal carboxyl groups of diverse peptides, and the residues lining the sides of the cleft bind to other residues in the ligand peptide through H bonding, similar to those between the strands in a β-sheet. Because such bonds are largely independent of peptide sequence, the binding site can accommodate a wide variety of C-terminal peptide sequences [72,80]. *C. albicans* also increases the availability of potential ligands by secreting Sap family proteases; their action will increase the number of exposed C-terminal sequences. In addition, Als adhesins bind to other partners through other types of interaction.

Als proteins also bind other peptide motifs, including those without free carboxyl groups. Als1 and Als5 each bind to a variety of heptapeptides even though the carboxy-terminal residue is covalently bound to large polyethylene glycol beads. There is a consensus sequence motif among most of these ligands: τϕ+, where τ is a residue common in turns, ϕ is an amino acid with a bulky hydrophobic side chain, and + is a cationic residue [81]. Als1 and Als5 show partially overlapping specificities among peptides with τϕ+ sequences. This motif is extremely common in proteins but is often buried in the interior of well-folded domains [81].

The many (5–35) tandem repeats in Als proteins (as well as *S. cerevisiae* Flo lectins) show highly hydrophobic surfaces. These small β-sheet domains unfold under shear stress to expose long unstructured peptides with high hydrophobicity. This part of the Als sequences mediates adhesion to plastic surfaces as well as to fibronectin [82,83].

In all Als proteins, there is a conserved sequence with high propensity to form amyloid-like cross-β structures. This sequence forms the interface between the Ig-like invasion domains and the highly conserved T region. Under very low shear stress, this region is exposed to solvent, and forms amyloid-like plaques or nanodomains composed of dozens or hundreds of Als adhesin molecules on the cell surface (Figure 7). Because of the broad binding specificity of Als adhesins, this clustering leads to multipoint attachment of Als-expressing cells to other cells or other substrates. The resulting cell-cell or cell-substrate binding is extremely strong and shear resistant. Amyloid-like cross-β bonds also form *between cells*, an adhesion necessary for the formation and maintenance of robust biofilms [84,85,86]. These intercellular bonds can only form between cells expressing the same amyloid core sequence, and thus they only form bonds between cells that express Als proteins. This strong and specific interaction gives biofilms physical cohesion and also excludes Als non-expressing heterologous cells from the center of biofilms. There is similar cross-β bonding mediated by high amyloid potential sequences in *C. albicans* Pga59 and in Flo adhesins of *S. cerevisiae.* Furthermore, sequences with amyloid-forming ability are predicted in poorly structured regions in most known fungal adhesins [16,55,72,87,88]. So, the formation of cross-β aggregates is probably a common property of the primary adhesins.

Thus, Als adhesins show a broad variety of binding specificities and surface sites. These include a strong binding site for diverse C-terminal peptides, unknown sites for binding τϕ+ sequences that have no free C-terminus, non-specific hydrophobic effect interactions, and cross-β bond formation between identical sequences. Furthermore, Als1 can bind to fucose-containing glycans, and it also anchors enolase to the cell wall [40]. Conversely, Als proteins are triggering ligands for host proteins including macrophage, T-cell, and epithelial receptors that shape innate and acquired immune responses during commensal and infectious interactions with mammalian hosts.

### 9.2. Unexpected Functions of Als Family Adhesins

Als proteins have diverse cellular functions that may or may not be related to their diverse adhesin activities. Surprisingly, Als1 plays a role in maintaining cell size [89], with deletants showing an increased fraction of small cells. The mechanisms behind this activity are unknown. More conventionally, Als adhesins mediate biofilm formation in *C. albicans,* as well as the binding and inclusion of bacteria and other fungi in mixed species biofilms [90]. Als adhesins bind multiple proteins on the surface of human oral epithelia, where Als3 interacts with the gC1qR receptor, leading to autophosphorylation of the epithelial growth factor receptor EGFR, and subsequent endocytosis of the fungus [63,91,92]. At the same time, Als3-expressing yeasts can promote adhesion, invasion, and disseminated infections of *Staphylococcus aureus* [93]. As cell surface antigens, Als proteins are modulators of innate immune responses, including blocking NK (natural killer T cell) cell-mediated killing of fungi. Als6, Als7, and the Als9-2 allele bind to the NK cell checkpoint receptor TIGIT to limit NK cell response. This interaction leads to increased fungal burden and host mortality [94]. Another immune invasion strategy that is triggered by Als proteins is the ability to bind the innate immune pattern receptor Serum Amyloid P component (SAP). SAP binds to amyloid core sequences in *C. albicans*, which masks the yeast against macrophage phagocytosis, and also skews macrophages to the non-inflammatory M2 state [74,76].

## 10. The Remarkable Similarity of Bacterial MSCRAMMs and Fungal Als Adhesins

The N-terminal region Ig-like domains region of Als adhesins are structurally similar to the Ig-like domains of MSCRAMMs and Als3 (Figure 8) [48,72,80,95]. Als proteins bind C-terminal peptides in the cleft between the domain, like the binding trench in MSCRAMMs. In each of these adhesin families, the Ig-like domains are followed in sequences by tandem β-sheet domains, a long unstructured stalk region that holds the adhesin domains away from the cell surface, and a covalent bond to the microbe wall. There is the sortase-dependent transpeptidylation of MSCRAMMs to the wall peptidoglycan, and in the fungi there is the GPI-dependent transglycosylation of adhesins to the wall β-glucans. In each family, the peptide immediately C-terminal to Ig-like domain 2 has a non-structured to structured transition: in MSCRAMMs it is the DLL lock and latch mechanism. In Als proteins this peptide forms amyloid-like cross-β bonds that cluster the adhesins to increase avidity. In each family, at least some members can form these cross-β amyloids between cells.

The similarities extend to multifunctional properties of these adhesins. Both families mediate microbial binding to abiotic and biological substrata, including indwelling medical devices. Each family shows broad binding to mammalian ECM components, and each family includes members that act as epithelial invasins. In each family, there are secondary binding sites for other ligands, and these secondary sites are used to form homophilic bonds. These bonds cluster the adhesins on the cell surface to greatly increase the avidity of binding to complex substrata. These bonds are amyloid-like in Als proteins and MSCRAMM properties are consistent with that model. Als and MSCRAMMs also form amyloid-like cross-β bonds between kindred cells to strengthen biofilms under shear stress [96,97]. Both adhesin families show flow-dependent strengthening of the adhesive bonds (catch bonding), and in Als proteins this is due to the shear-dependent unfolding of the amyloid-containing peptide. The structures of MSCRAMMs and Als adhesins therefore appear to be similar, and they mediate similar multifunctional activities. Thus, there are adhesins with similar multifunctionality in bacteria and eukaryotic fungi.

## 11. Conclusions

Microbial adhesins are more than just ligand-binding proteins, and as we have documented, multifunctionality is a common property. These activities include three basic activities: cell-surface association, specific binding to a ligand or protein partner, and activity as a cell surface marker. In addition, there are often multiple binding specificities at multiple binding sites. These diverse interactions lead to a variety of outcomes. For example, Als family and MSCRAMM family activities lead to host interactions, biofilm formation, and the alteration of gene expression patterns and physiology of the microbes. This multifunctionality illustrates the principle that in protein structure there is opportunity for multiple interactions on any protein surface. Thus, we commonly see protein aggregation in quaternary interactions (e.g., the pyruvate dehydrogenase complex and the DNA polymerase nanomachine), macromolecular assemblies such as ribosomes and chromosomal spindles, and aggregation to form membrane-less organelles. We have also mentioned that *C. albicans* enolase has evolved to have transaminase activity. Other examples of multifunctionality in fungal proteins are the subject of other articles in this issue. These concepts illustrate that evolution often takes advantage of opportunities to reuse molecules for new traits. So, we may find that multiple functions are the norm for proteins, and that monofunctional proteins become rarer as we discover more about life on earth.

## Figures and Tables

**Figure 1 jof-09-00419-f001:**
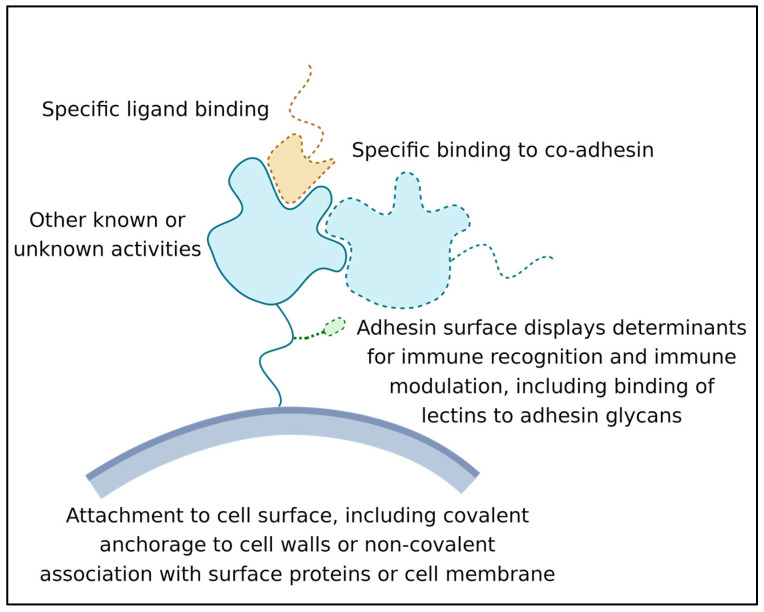
Some general features and activities of microbial adhesins. Adhesins (blue with a green glycosylation) are bound to cell wall (grey). The ligand is shown in tan.

**Figure 2 jof-09-00419-f002:**
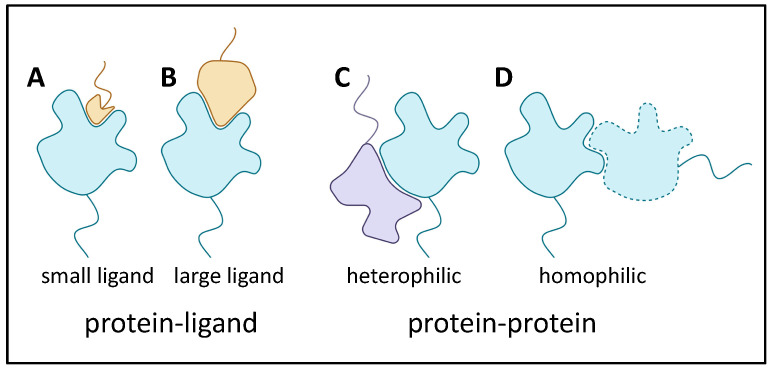
Adhesin binding to ligands and binding partners. Colors are the same Figure 1, with a heterophilic binding partner (lavender in C).

**Figure 3 jof-09-00419-f003:**
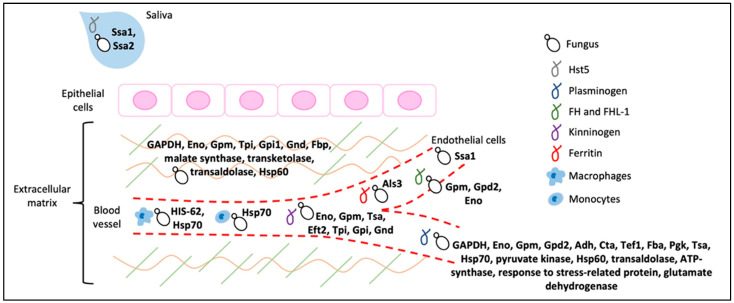
*C. albicans* cytoplasmic adhesins. Adhesins are bolded, and their binding partners are cartooned. Reprinted with permission from [34]: 2022, Hector Mora-Montes.

**Figure 4 jof-09-00419-f004:**
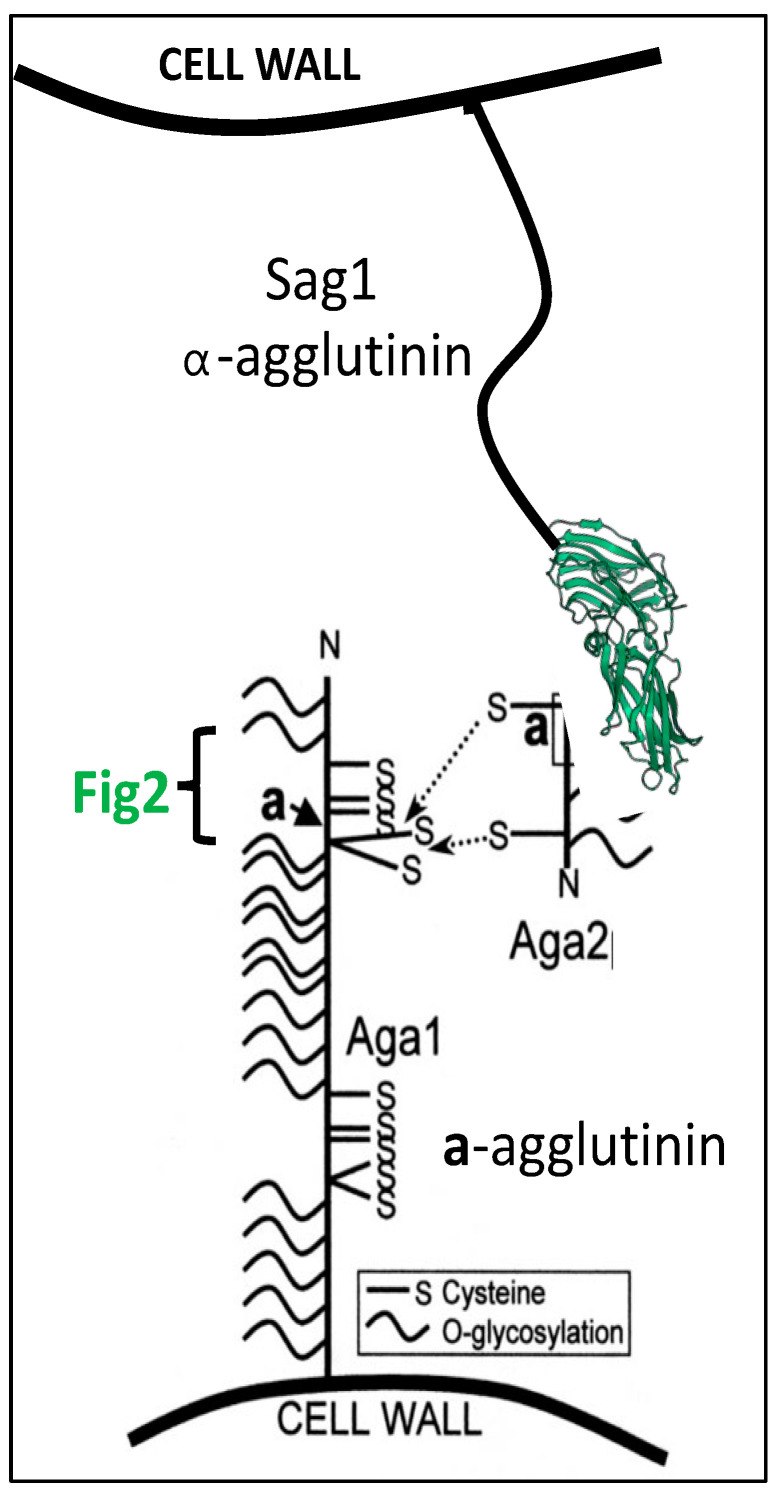
*S. cerevisiae* sexual agglutinins. α-Agglutinin (Sag1) expressed on α-cells is shown at the **top**, binding to the C-terminal sequence of its ligand Aga2. Aga2 is a disulfide-bonded subunit of **a**-agglutinin expressed on the surface of **a**-cells (bottom). The **a**-agglutinin Aga1 subunit is expressed on both mating types and can also bind to the adhesin Fig2 within the bracketed region. Adapted with permission from [61]: 2001, Peter Lipke.

**Figure 5 jof-09-00419-f005:**
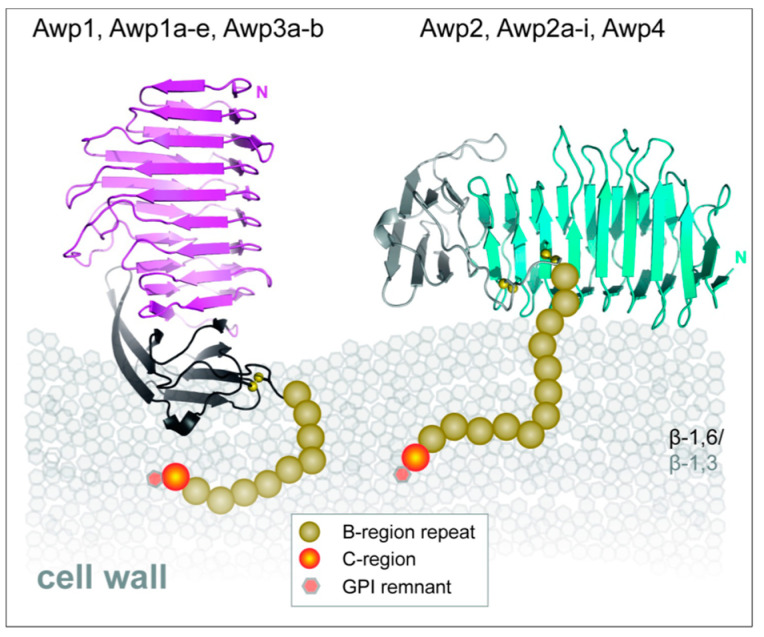
*C. glabrata* Awp adhesins. These adhesins have cylindrical β-barrel adhesive domains. Image reused with permission from [70]: 2021, Lars-Oliver Essen.

**Figure 6 jof-09-00419-f006:**
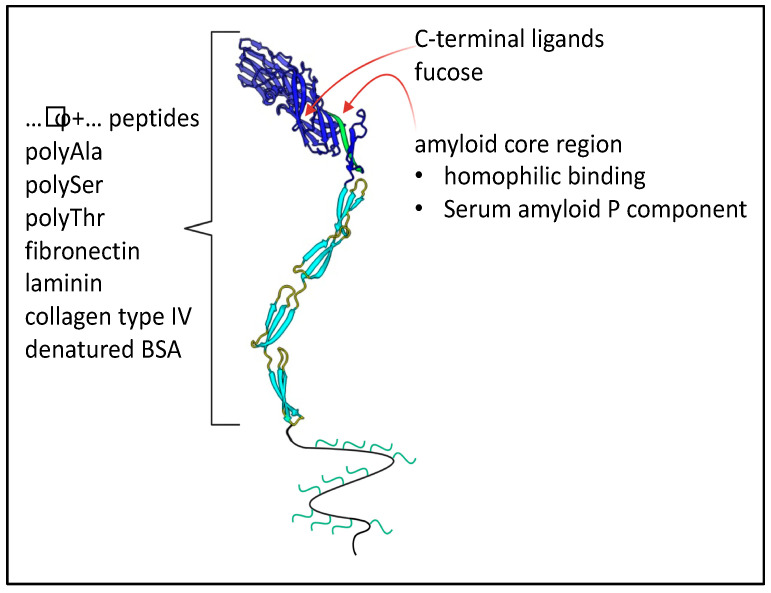
Binding partners of *C. albicans* Als adhesins. The model shows the two N-terminal Ig-like domains in dark blue, with the binding cleft between them. Three tandem repeats in the T region and two of many repeats in the TR regions are shown in light blue. The unstructured C-terminal stalk region is shown as a wavy line with attached glycans in green. This region typically consists of 600–1000 residues.

**Figure 7 jof-09-00419-f007:**
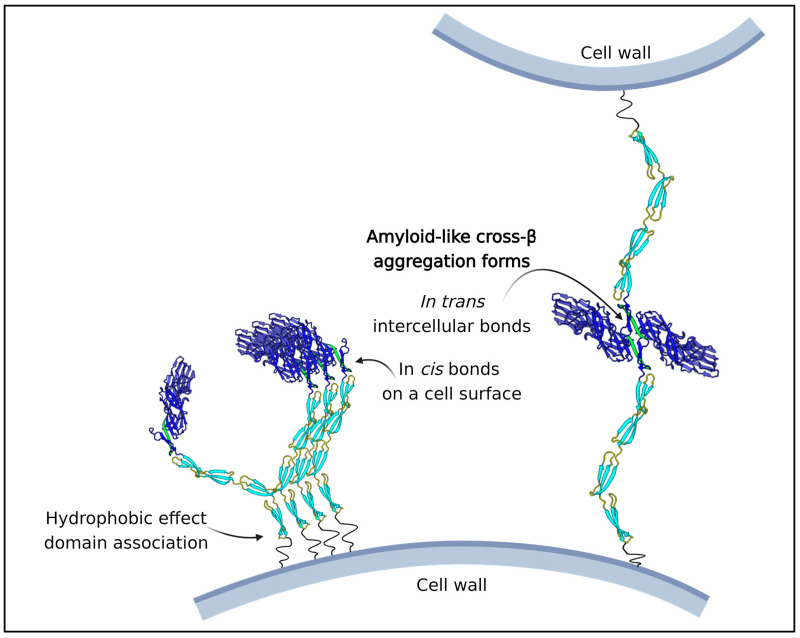
Roles of the amyloid core in the T region and the hydrophobic tandem repeats in aggregation of Als adhesins. The amyloid core regions (green) are exposed under low shear stress, and then form cross-β bonds between adhesins on the surface of a cell (in *cis)*, and between two cells (in *trans*). The tandem repeats (light blue) also cluster the adhesins through hydrophobic interactions (left) [82]. The long unstructured stalk regions (black squiggle lines) add the extension and flexibility that allow these interactions to occur.

**Figure 8 jof-09-00419-f008:**
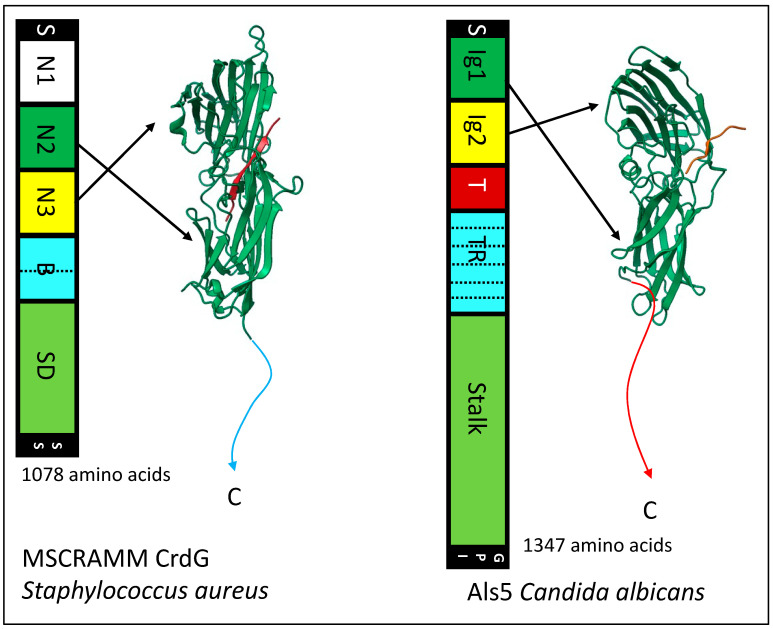
Similarities between a bacterial MSCRAMM and a fungal Als adhesin. In the maps, regions with analogous structures are similarly colored: N-terminal secretion signals (S) and C-terminal wall anchorage signals (SS, bacterial sortase; GPI glycosyl phosphatidyl inositol) are black, Ig-like domains are green and yellow, tandem repeats are light blue, and low-complexity stalk regions are light green (SD, (SerAsp)_n_ repeats). The structural models show the Ig-like domains with the ligand peptides in orange. The CrdG structure is from pdb file 1R17 and the Als3 structure from file 4LEB.

## Data Availability

All data is contained within the manuscript.

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
