# Peer review of "Sticking to the Subject: Multifunctionality in Microbial Adhesins"

_jof, 2023, doi:10.3390/jof9040419_

Round 1

Reviewer 1 Report

1.       This is a very interesting review describing the structure and functions of many  fungal adhesins and a few bacterial adhesins.

2.       JOF is journal dedicated to fungi it is fine to include bacterial adhesins in this review. However, the authors describre3 groups of bacterial adhesins: 4.1 Bacillus subtilis` 4.2 Pili adhesins` 4.3 MSCRAMMS. Though a comparison indeed been made between the fungal adhesins  and the 3rd group of bacterial adhesin. Therefore, a comparison should also be done for first 2 groups of bacterial adhesins to fungal adhesins as well.

3.       The authors describe many families of fungal adhesins. However, they did not mention the integrin-like protein (ILP) which is important family of fungal adhesins and should be included in this review.

4.       The C. albicans ALS adhesins have both “social” (aggregative) and “antisocial” (pathogenic) functions - These definitions which is common in the literature should also be specified in this article.

5.        The authors used a large number of abbreviations like ECM, GPI ALS and more, as without giving details of their full names when they are first mentioned, please correct.

6.       I recommend that the authors should give an illustration of the FLO adhesin that appears in section 5, line 218, just as they gave illustrations to the other adhesins.

7.       Line 292 please insert the names of the adhesin families I assume EPA and AWP.

8.        The quality of the illustration in Figure 5 is poor and blurry and needs improvement so that the printout looks clear.

9.       In line 314 the authors wrote that there are 8 ALS allelic genes however in lines 404 and 412 they mentioned ALS9. Please check whether it is not needed to correct to 9 ALS genes.

10.    Figure 8 shows an illustration of ALS5 and not of ALS9 as written in line 412, please correct.

11.   I recommend to add the following review reference - Richard A. Calderone and William A. Fonzi. Virulence factors of Candida albicans  Trends In Microbiology VOLUME 9, ISSUE 7, P327-335, , 2001

Reviewer 2 Report

This beautiful review summarized the structure-function relationships of fungal and bacterial adhesins. The review is well organized and written; no weakness is spotted. It will be a great addition to the literature.

Author Response

We thank the reviewer for their comment

Reviewer 3 Report

This review focuses on the multifunctionality of professional and moonlighting microbial adhesins. It makes a very good contribution to the field. Sections 5-10 are very well written. Sections 1-4 require some improvement.

Comments:

While the English language is generally of a high standard, there are grammatically errors throughout the abstract and sections 1-4 that need correction.

Throughout sections 1 and 2 there are insufficient citations, and there are many statements for which references should be added.

l.30: Generally the term “active site” only applies to enzymes. The term “functional domain” may be more appropriate throughout this paragraph.

p.52: Adhesins are generally defined not only as proteins that mediate cell-cell interactions, but those that mediate attachment of cells to any surface. The text should be modified to reflect this and to describe the specific group of cell-cell adhesins that are the focus of this review.

l. 136. Receptors generally refer to molecules on host cells that bind ligands on microbes. Therefore the following sentence should be removed or re-written: “In fact, the standard ligand-receptor terminology is often inappropriate, because both binders are adhesins, and either partner can be called the receptor.”

l.175: Change “strain” to “species”. Delete “the idea of a complete catalog is scary indeed”.

l.331 and 332. There are 2 authors of this review, therefore remove “I”, and “my” from these sentences.

Fig 1 and 2: The figure legends should include a description of what each colour represents in the figure.

Figures 3, 4, 5 are reproduced from other publications. They should be removed, as they do not make a large contribution to the reader’s understanding of the review material.

Fig 8: Change “Staph” to “Staphylococcus”
